# EFFICIENT ESTIMATORS FOR HEAVY-TAILED MACHINE LEARNING

## ABSTRACT

A dramatic improvement in data collection technologies has aided in procuring massive amounts of unstructured and heterogeneous datasets. This has consequently led to a prevalence of heavy-tailed distributions across a broad range of tasks in machine learning. In this work, we perform thorough empirical studies to show that modern machine learning models such as generative adversarial networks and invertible flow models are plagued with such ill-behaved distributions during the phase of training them. To alleviate this problem, we develop a computationally-efficient estimator for mean estimation with provable guarantees which can handle such ill-behaved distributions. We provide specific consequences of our theory for supervised learning tasks such as linear regression and generalized linear models. Furthermore, we study the performance of our algorithm on synthetic tasks and real-world experiments and show that our methods convincingly outperform a variety of practical baselines.

## 1 INTRODUCTION

Existing estimators in machine learning are largely designed for "thin-tailed" data, such as those coming from a Gaussian distribution. Past work in statistical estimation has given sufficient evidence that in the absence of these "thin-tails", classical estimators based on minimizing the empirical error perform poorly (Catoni, 2012; Lugosi et al., 2019). Theoretical guarantees for methods commonly used in machine learning usually place assumptions on the tails of the underlying distributions that are analyzed. For instance, rates of convergences proven for a variety of stochastic optimization procedures assume that the distribution of gradients have bounded variance (for e.g., Zou et al. (2018)) or in some cases are sub-Gaussian (for e.g., Li & Orabona (2019)). Thus, these guarantees are no longer applicable for heavy-tailed gradient distributions.

From a practical point of view however, this is a less than desirable state of affairs: heavy-tailed distributions are ubiquitous in a variety of fields including large scale biological datasets and financial datasets among others (Fan et al., 2016; Zhou et al., 2017; Fan et al., 2017). While this may be argued as just artifacts of the domain, recent work has found interesting evidence of heavy-tailed distributions in the intermediate outputs of machine learning algorithms. Specifically, recent work by Simsekli et al. (2019) and Zhang et al. (2019) have provided empirical evidence about the existence of such heavy-tailed distributions, especially during neural network training for supervised learning tasks.

Following these empirical analyses of Simsekli et al. (2019) and Zhang et al. (2019), we look for sources of heavy-tailed gradients arising during the training of modern generative model based unsupervised learning tasks as well. In our preliminary investigation, we noticed that the distribution of gradient norms i.e., $\|g_t\|_2$ are indeed heavy-tailed. These are showcased in Figure 1; Figures 1a and 1b show the distribution of gradient norms obtained while training the generator of a `DCGAN` (Radford et al., 2015) and `Real-NVP` (Dinh et al., 2016) on the CIFAR-10 dataset, respectively. These distributions are noticeably heavy-tailed, especially when juxtaposed with those obtained from a Gaussian distribution (Figure 1c). We discuss more about the empirical setup in Section 5.2.

Interestingly, in all the supervised and unsupervised machine learning problems discussed above, we merely need to compute expectations of these varied random heavy-tailed quantities. For instance, mini-batch gradient descent involves aggregating a batch of gradients pertaining to each sample in

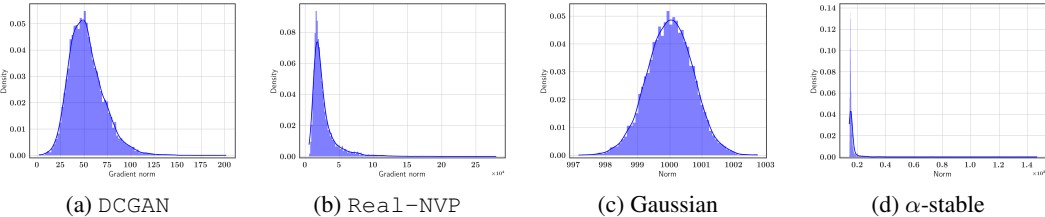

Figure 1: Distribution of sampled gradient norms while training `DCGAN` (a) and `Real-NVP` (b) on the CIFAR-10 dataset. (c) Distribution of norms of Gaussian random vectors and (d) Distribution of norms of $\alpha$-stable random vectors with $\alpha = 1.95$. X-axis: norm, Y-axis: Density.

the mini-batch. Typically, this aggregation is performed by considering the sample mean, and this is a reasonable choice due to its simplicity as an estimate of the expectation of the random gradient.

For computing the mean of such heavy-tailed gradient distributions, the sample mean however is highly sub-optimal. This is because sample mean estimates are greatly skewed by samples on the tail. Thus gradient estimates using these sub-optimal sample means of gradients do not necessarily point in the right direction leading to bad solutions, prolonged training time, or a mixture of both. Thus, a critical requirement for training of modern machine learning models is a scalable estimation of the mean of a heavy-tailed random vector. Note that such computations of mean of sample gradients are done in each iteration of (stochastic) gradient descent, so that we require that the heavy-tailed mean estimation be extremely scalable, yet with strong guarantees. Note that once we have such a scalable heavy-tailed mean estimator, we could simply use it to compute robust gradient estimates Prasad et al. (2020) , and learn generic statistical models.

We summarize our contributions as follows:

- We extend recent analyses of heavy-tailed behavior in machine learning, and provide novel empirical evidence of heavy-tailed gradients while training modern generative models such as generative adversarial networks (GANs) and invertible flow models.

- To combat the issue of aggregating gradient samples from a heavy-tailed distribution, we propose a practical and easy-to-implement algorithm for heavy-tailed mean estimation with provable guarantees on the error of the estimate.

- We use the proposed mean estimator to compute robust gradient estimates, which allows us to learn generalized linear models in the heavy-tailed setting, with strong guarantees on the estimation errors.

- Finally, we propose a heuristic approximation of the mean estimation algorithm, which scales to random vectors with millions of variables. Accordingly, we use this heuristic to compute robust gradients of large-scale deep learning models with millions of parameters. We show that training with this heuristic outperforms a variety of practical baselines.

**Notation and other definitions.** Let $x$ be a random vector with mean $\mu$. We say that the $x$ has bounded $2k$-moments if for all $v \in \mathcal{S}^{p-1}$ (unit ball), $\mathbb{E}[(v^T(x - \mu))^{2k}] \leq C_{2k} \left( \mathbb{E}[(v^T(x - \mu))^2] \right)^k$. Throughout the paper, we use $c, c_1, c_2, \ldots, C, C_1, C_2, \ldots$ to denote positive universal constants.

## 2 EFFICIENT AND PRACTICAL MEAN ESTIMATION

We begin by formalizing the notion of heavy-tailed distributions.

**Definition 1** (Heavy-Tailed Distribution (Resnick, 2007)). *A non-negative random variable $X$ is called heavy-tailed if the tail probability $P(X > t)$ is asymptotically proportional to $t^{-\alpha^*}$, where $\alpha^*$ is a positive constant called the tail index of $X$.*

Intuitively, this definition states that if the tail of the distribution $P(X > t)$ decreases at a rate slower that $e^{-t}$, then the distribution is heavy-tailed. An interesting consequence of this definition is the non-existence of higher order moments. Specifically, one can show that the quantity $\mathbb{E}[X^\alpha]$ is finite for any $\alpha$ if and only if $\alpha < \alpha^*$ and $X$ is a heavy-tailed random variable with tail index $\alpha^*$. In recent statistical estimation literature (for e.g., Minsker (2015); Hopkins (2018); Lugosi & Mendelson (2019)), heavy-tailed distributions are defined by the absence of finite higher order moments.

In the heavy-tailed mean estimation task, we observe $n$ samples $x_1, \ldots, x_n$ drawn independently from a distribution $P$ where $x_i \in \mathbb{R}^p$, which is only assumed to have finite low-order moments, therefore heavy-tailed. The goal of past work (Catoni, 2012; Minsker, 2015; Lugosi et al., 2019; Catoni & Giulini, 2017) has been to design an estimator $\widehat{\theta}_n$ of the true mean $\mu$ of $P$ which has a small $\ell_2$-error with high-probability.

As a benchmark for estimators in the heavy-tailed model, we observe that when $P$ is the multivariate normal (or equivalently a sub-Gaussian) distribution with mean $\mu$ and covariance $\Sigma$, the sample mean $\widehat{\mu}_n = 1/n \sum_i x_i$ satisfies, with probability at least $1 - \delta$ [1]:

$$\|\widehat{\mu}_n - \mu\|_2 \lesssim \sqrt{\frac{\operatorname{trace}(\Sigma)}{n}} + \sqrt{\frac{\|\Sigma\|_2 \log(1/\delta)}{n}} \stackrel{\text{def}}{=} \operatorname{OPT}_{n,\Sigma,\delta} \tag{1}$$

Seminal work by Catoni (2012) showed that the sample mean is extremely sub-optimal, while more recent work by Lugosi et al. (2019) showed that the sub-Gaussian error bound is achievable while *only assuming that $P$ has finite variance* i.e., $2^{nd}$ moment. In the multivariate setting, Minsker (2015) showed that the extremely practical geometric-median-of-means estimator (GMOM) achieves a sub-optimal error bound by showing that with probability at least $1 - \delta$:

$$\|\widehat{\theta}_{\text{MOM},\delta} - \mu\|_2 \lesssim \sqrt{\frac{\operatorname{trace}(\Sigma)\log(1/\delta)}{n}}. \tag{2}$$

Computationally intractable estimators that truly achieve the sub-Gaussian error bound were proposed by Lugosi et al. (2019); Catoni & Giulini (2017). Hopkins (2018) and later Cherapanamjeri et al. (2019) developed a sum-of-squares based relaxation of the estimator in Lugosi et al. (2019), thereby giving a polynomial time algorithm which achieves optimal rates. More recent work has studied the problem of mean estimation, focusing on constructing theoretically fast polynomial time estimators (Dong et al., 2019; Lugosi & Mendelson, 2019; Diakonikolas & Kane, 2019; Lei et al., 2020; Lecué & Depersin, 2019). However, these estimators have several hyperparameters, which require to be tuned for practice, making them impractical.

Now, we present our algorithm Filterpd for heavy-tailed mean estimation, and is formally stated as Algorithm 1. It proceeds in an iterative fashion, by (1) computing the leading eigenvector of the empirical covariance matrix (Step (3)), (2) projecting points along this leading eigenvector (Step (4)), and (3) randomly sampling points based on their projection scores (Step (5) and (6)). This procedure is repeated for a fixed number of steps.

---

**Algorithm 1** Filterpd - Heavy Tailed Mean Estimator

**Require:** Samples $S = \{z_i\}_{i=1}^n$, Iterations $T^*$
1: **for** $t = 1$ **to** $T^*$ **do**
2:      Compute $\widehat{\theta}_S = \frac{1}{|S|}\sum_{i=1}^{|S|} z_i$ and $\widehat{\Sigma}_S = \frac{1}{|S|}\sum_{i=1}^{|S|}(z_i - \widehat{\theta}_S)^{\otimes 2}$
3:      Let $v$ be the leading eigenvector of $\widehat{\Sigma}_S$
4:      For each $z_i$, let $\tau_i \stackrel{\text{def}}{=} \left(v^T(z_i - \widehat{\theta}_S)\right)^2$ be its score.
5:      Sample a point $z \sim S$ according to $\Pr(z_i) \propto \tau_i$
6:      Remove sample $z$ from $S$ *i.e.* $S = S \setminus \{z\}$
7: **end for**
8: **return** $\frac{1}{|S|}\sum_{i=1}^{|S|} z_i$

---

Our proposed algorithm is primarily based on the SVD-based filtering algorithm, which has appeared in different forms (Klivans et al., 2009; Awasthi et al., 2014) and was recently reused in Diakonikolas et al. (2016; 2017) for adversarial mean estimation. For instance, the algorithm in Diakonikolas et al. (2017) follows a similar procedure, but remove a subset of points at a step depending on magnitude of the projection score.

Our first main result is presented as follows:

**Theorem 1.** *Suppose $\{z_i\}_{i=1}^n \sim P$ with $z_i \in \mathbb{R}^p$ for all i, where $P$ has bounded $4^{th}$ moment and $n$ satisfies*

$$n \geq C r^2(\Sigma) \frac{\log^2(p/\delta)}{\log(1/\delta)}, \qquad r(\Sigma) \stackrel{\text{def}}{=} \frac{\operatorname{trace}(\Sigma)}{\|\Sigma\|_2} \tag{3}$$

---

[1] Here and throughout our paper we use the notation $\lesssim$ to denote an inequality with universal constants dropped for conciseness.

*Then,* Filterpd *when instantiated for* $T^* = \lceil C \log(1/\delta) \rceil$ *steps returns an estimate* $\widehat{\theta}_\delta$ *which satisfies with probability at least* $1 - 4\delta$, $\delta \in (0, 0.25)$:

$$\|\widehat{\theta}_\delta - \mu\|_2 \lesssim \mathrm{OPT}_{n,\Sigma,\delta}$$

**Remarks:** Theorem 1 shows that when $n$ is sufficiently large, Filterpd returns a mean estimate that achieves the *optimal sub-Gaussian deviation bound*. This algorithm is also extremely practical as compared to existing algorithms (for e.g., Lei et al. (2020)) and engenders development of scalable variants, which we later describe in Section 4. We defer the proofs of Theorem 1 to the appendix.

## 2.1 PROVABLE ALGORITHMS FOR GENERALIZED LINEAR MODELS

At this stage, with an optimal mean estimator in hand, we explore its consequences for general supervised learning tasks, namely linear regression and generalized linear models. The goal is to design efficient estimators which work well in the presence of heavy-tailed data. To this end, we borrow the *robust gradient* framework in Prasad et al. (2020) and present it in Algorithm 2. In particular, RGD − Filterpd proceeds by passing the gradients at the current iterate $\theta^t$ through Filterpd in Step (5).

Note that Prasad et al. (2020) used a similar algorithm in their work, but used GMOM (Minsker, 2015) as their mean estimator, which led to weaker results in the heavy-tailed setting. We show that using Filterpd as the mean estimator automatically results in better bounds. The proofs for the technical results appearing henceforth can be found in the appendix.

---

**Algorithm 2** RGD − Filterpd
- Robust Gradient Descent (Prasad et al., 2020)

---

**Require:** Data $\{z_i\}_{i=1}^n$, Loss Function $\bar{\mathcal{L}}$.
**Require:** Step size $\eta$, Number of Iterations $T$, Confidence $\delta$.
**Require:** Initialization $\theta^0$ and constraint set $\Theta$.
1: Split samples into $T$ subsets $\{\mathcal{Z}_t\}_{t=1}^T$ of size $\widetilde{n} = \lfloor n/T \rfloor$
2: Set $T^* = C \log(T/\delta)$
3: **for** $t = 1$ to $T$ **do**
4:     Obtain $S_t = \{\nabla \bar{\mathcal{L}}(\theta^{t-1}; z_i) : z_i \in \mathcal{Z}_t\}$
5:     Let $g^t = \mathsf{Filterpd}(S_t, T^*)$
6:     Update $\theta^t = \underset{\theta \in \Theta}{\arg\min} \|\theta - (\theta^{t-1} - \eta g^t)\|_2^2$
7: **end for**
8: **return** $\{\theta_t\}_{t=1}^T$

---

In this setting, we observe pairs of samples $\{(x_1, y_1), \ldots (x_n, y_n)\}$, where each $(x_i, y_i) \in \mathbb{R}^p \times \mathbb{R}$. We assume that the $(x, y)$ pairs sampled from the true distribution $P$ are linked via a linear model:

$$y = x^T \theta^* + w, \qquad (4)$$

where $w$ is drawn from a zero-mean distribution with bounded $4^{th}$ moment with variance $\sigma^2$. We suppose that under $P$ the covariates $x \in \mathbb{R}^p$, have mean $\mathbf{0}$, covariance $\Sigma_x$ satisfying $\tau_\ell \mathcal{I}_p \preceq \Sigma_x \preceq \tau_u \mathcal{I}_p$ and bounded $8^{th}$ moment.

**Theorem 2.** *Consider the statistical model in* (4). *Suppose we observe* $n$ *pairs of samples, where* $n$ *satisfies*

$$n \geq C_1 p^2 \frac{T \log^2(pT/\delta)}{\log(T/\delta)}. \qquad (5)$$

RGD − Filterpd *when initialized at* $\theta^0$ *with step size* $\eta = 2/(\tau_u + \tau_\ell)$ *and confidence parameter* $\delta$ *returns iterates* $\{\widehat{\theta}^t\}_{t=1}^T$ *which with probability at least* $1 - \delta$ *satisfy:*

$$\|\widehat{\theta}^t - \theta^*\|_2 \leq \kappa^t \|\theta^* - \theta^0\|_2 + \frac{C_2 \sigma}{1 - \kappa} \left( \sqrt{\frac{\mathrm{trace}\,(\Sigma_x)}{n/T}} \right) + \frac{C_2 \sigma}{1 - \kappa} \left( \sqrt{\frac{\|\Sigma_x\|_2 \log(T/\delta)}{n/T}} \right) \qquad (6)$$

*for some contraction parameter* $\kappa < 1$.

**Remarks:** For an appropriately chosen $T$, we achieve the best known result for heavy-tailed linear regression, improving on the previously best known rate by Hsu & Sabato (2016). A detailed exposition with the form of $T$ and detailed comparisons to other work is presented in the appendix due to space constraints.

We also note that RGD − Filterpd is also effective for generalized linear models (GLMs) such as logistic regression in the heavy-tailed setting. We extend guarantees for these statistical models in the appendix (Sections B.3 and B.4) due to space constraints.

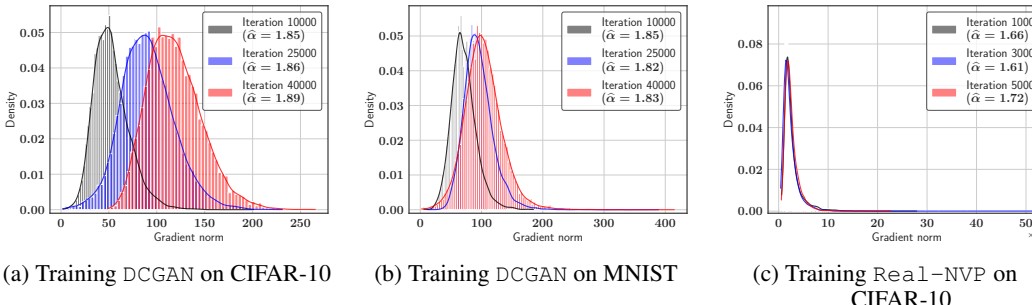

(a) Training `DCGAN` on CIFAR-10    (b) Training `DCGAN` on MNIST    (c) Training `Real-NVP` on CIFAR-10

Figure 2: Variation of gradient distributions across different iterations for different models and datasets. (a) and (b) show the variation of generator gradient norms over iterations for `DCGAN` and (c) shows the variation of the complete gradient norms over iterations for `Real-NVP`. X-axis: Gradient norm, Y-axis: Density

## 3 EXISTENCE OF HEAVY-TAILS IN DEEP GENERATIVE MODELS

In this section, we elaborate on the preliminary study discussed in Section 1. The evidence of heavy-tailed gradient distributions observed in `DCGAN` and `Real-NVP` suggests that the issue of heavy-tailedness is more pervasive than thought to be. While recent analyses have only considered supervised learning models focusing on image classification (Simsekli et al., 2019) and on attention models (Zhang et al., 2019), they miss out a large family of models that benefit from gradient estimates, namely probabilistic models (Mohamed et al., 2019). Examples of probabilistic models are generative adversarial networks (GANs) (Goodfellow et al., 2014) and invertible flow models (Dinh et al., 2014), and these are the models we have considered in our study.

Extending on the study from earlier, we investigate the variation in the norms of the gradients across iterations while training. Zhang et al. (2019) had previously identified the effect of different datasets for their supervised learning setup. Therefore, we also consider an additional dataset in our study with GANs to observe any effects that the data distribution could have on this distribution. The plots are showcased in Figure 2.

To quantitatively assess the heavy-tailedness of these gradient norm distributions, we use an estimator to measure the heavy-tailedness, which assumes that the underlying distribution is a strictly $\alpha$-stable distribution. We use the $\alpha$-index estimator proposed by Mohammadi et al. (2015) and used in Simsekli et al. (2019). While there are some drawbacks of using this estimator, such as the requirement of symmetric distributions and the assumption that the distribution is $\alpha$-stable, this estimator provides a rough idea about how heavy the tail of a distribution is given samples from that distribution. Formally, given $n$ samples $\{X_i\}_{i=1}^n$ and natural numbers $K_1, K_2$ such that $K_1 \cdot K_2 = n$, the $\alpha$-index estimate $\widehat{\alpha}$ is computed as follows

$$\widehat{\alpha} = \log(K_2) \left( \frac{1}{n} \sum_{i=1}^n \log(|X_i|) - \frac{1}{K_2} \sum_{i=1}^{K_2} \log(|Y_i|) \right)^{-1} \qquad Y_i = \frac{1}{K_1} \sum_{j=1}^{K_1} X_{j+(i-1)K_2}$$

An $\alpha$-index close to 2 indicates that the samples are closer to being Gaussian, and a lower $\alpha$-index is indicative of heavy-tailedness. In the legend of each plot in Figure 2, we specify the estimated $\alpha$-index for the gradient norm distributions. An iteration-wise variation is presented in the appendix (Section C), along with an alternate method to quantify heavy-tailedness.

With the presented evidence about the existence of heavy-tailed distributions in certain deep generative models, the use of a heavy-tailed mean estimator could be beneficial. However, Filterpd requires to be scaled to work for models with millions of parameters, and for sample sizes that are much smaller than the number of parameters.

## 4 Streaming − Filterpd: A HEURISTIC FOR LARGE-SCALE MODELS

Modern large-scale machine learning models have millions of parameters, and hence, are generally trained using stochastic (or mini-batch) gradient descent (Robbins & Monro, 1951) in practice. In such a setting, we cannot directly use Filterpd to aggregate gradients. This is because when the

---

**Algorithm 3** Streaming − Filterpd

---

**Require:** History parameter $\alpha$, Iterations $T$, Discard parameter $d$, Initialization $\theta_0$, Optimizer ALG
1: **for** $t = 0$ **to** $T - 1$ **do**
2:     Obtain gradient samples $\{g_t^{(i)}\}_{i=1}^n$ of the objective at $\theta_t$.
3:     **if** $t = 0$ **then**
4:         Compute $m_t = \dfrac{1}{n}\sum_{i=1}^n g_t^{(i)}$ and $C_t = \dfrac{1}{n}\sum_{i=1}^n (g_t^{(i)} - m_t)^{\otimes 2}$.        $\left(v^{\otimes 2} \overset{\text{def}}{=} vv^T\right)$
5:     **else**
6:         Update $m_t = (1-\alpha)m_{t-1} + \dfrac{\alpha}{n}\sum_{i=1}^n g_t^{(i)}$, $C_t = (1-\alpha)\lambda_{t-1}v_{t-1}^{\otimes 2} + \dfrac{\alpha}{n-1}\sum_{i=1}^n (g_t^{(i)} - m_t)^{\otimes 2}$
7:     **end if**
8:     Construct $B_t$ whose columns are $\{\sqrt{(1-\alpha)\lambda_{t-1}}\,v_{t-1}\} \cup \{\sqrt{\alpha/n-1}(g_t^{(i)} - m_t)\}_{i=1}^n$
9:     Cache $\lambda_t$ and $v_t$ – the leading eigenpair of $C_t = B_t B_t^T$
10:    Calculate scores $\tau_i = (v_t^T(g_t^{(i)} - m_t))^2$ and discard the top-$d$ gradients ordered by scores
11:    Compute final estimate $G(\theta_t) = \dfrac{1}{n-d}\sum_{i=1}^{n-d} g_t^{(i)}$, obtain update $\theta_{t+1} = \mathsf{ALG}(G(\theta_t), \theta_t)$
12: **end for**
13: **return** $\theta_T$

---

mini-batch size $n$ is much smaller than the dimension $p$, the sample covariance matrix does not concentrate well around the true covariance matrix. Due to this, the leading eigenvector of the sample covariance matrix could be significantly different from that of the true covariance matrix. Moreover, it is not viable to run $T^*$ leading eigenvector computations for models with millions of parameters, since the additional time complexity of this operation is proportional to $p$ and is not as efficient as computing the sample mean.

We now look at approaches to mitigate the above issues and present the spirit of the approaches. The first issue is with regard to the concentration of sample covariance. For this, we intend on reusing previous gradient samples to improve concentration. However, the gradient samples from earlier iterations cannot be weighted equally as neither the distributions are the same nor are the samples independent. Thus, given gradient samples at the current iteration and the mean and covariance from the previous iteration, a potential solution could be to compute the current estimates of the samples statistics by an exponential average like so.

$$m_t = (1-\alpha)m_{t-1} + \alpha\widehat{\mu}_t \qquad\qquad C_t = (1-\alpha)C_{t-1} + \alpha\widehat{\Sigma}_t$$

where $\widehat{\mu}_t$ and $\widehat{\Sigma}_t$ are the mean and covariance of the gradient samples at iteration $t$ and $m_{t-1}$ and $C_{t-1}$ are the mean and covariance computed at the previous iteration. This solution, while promising, is not tractable, as it requires storing the complete covariance matrix. To circumvent this, we approximate $C_{t-1}$ by a rank-one matrix, which is given by the eigenpair $(\lambda_{t-1}, v_{t-1})$ of the covariance matrix $C_{t-1}$.

Recall that in Filterpd, we require to compute $T^*$ leading eigenpairs. To reduce computational costs, we propose running one leading eigenpair computation at every iteration. This leading eigenpair is used in two ways: 1) to compute "outlier scores", and 2) to compute the exponential weighted covariance in the next iteration as discussed above.

The heuristic is formally presented in Algorithm 3. The aforementioned exponential averaging is performed in Step (6). Step (8) is a computational trick used to avoid computing the complete covariance matrix $C_t$. The leading eigenpair and the scores of the gradient samples are computed in Steps (9) and (10) respectively. We discard points based on scores obtained using the computed eigenvector (the analogous steps in Filterpd are Steps (5) and (6)), and finally pass our robust aggregate to the optimization algorithm of choice in Step (11) – this allows us to leverage the benefits of certain optimization algorithms. We repeat this for $T$ iterations as done in standard training procedures.

The algorithm is memory efficient due to Steps (8) and (9). This is due to the construction of the matrix $B_t$ which only requires $\mathcal{O}(np)$ memory. From a computation standpoint, we only require the leading left singular vector of $B_t$, which takes $\mathcal{O}(np)$ time to run. In contrast, a naive implementation of this algorithm is not memory efficient since it would require computing a $p \times p$ matrix $C_t$

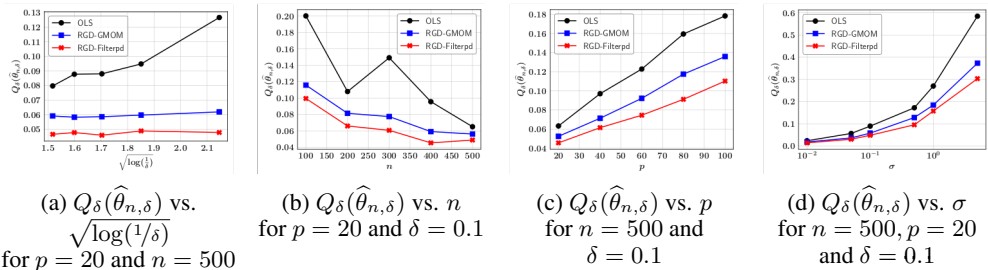

(a) $Q_\delta(\widehat{\theta}_{n,\delta})$ vs. $\sqrt{\log(1/\delta)}$ for $p = 20$ and $n = 500$

(b) $Q_\delta(\widehat{\theta}_{n,\delta})$ vs. $n$ for $p = 20$ and $\delta = 0.1$

(c) $Q_\delta(\widehat{\theta}_{n,\delta})$ vs. $p$ for $n = 500$ and $\delta = 0.1$

(d) $Q_\delta(\widehat{\theta}_{n,\delta})$ vs. $\sigma$ for $n = 500, p = 20$ and $\delta = 0.1$

Figure 3: Results for Heavy-Tailed Linear Regression. Smaller values for $Q_\delta(\widehat{\theta}_{n,\delta})$ are better. Legend: (red, cross) - $\mathsf{RGD - Filterpd}$, (blue, square) - $\mathsf{RGD - GMOM}$, (black, circle) - OLS

and computing the leading eigenvector which can take $\mathcal{O}(p^2)$ time. In our real-world experiments described in Section 5.2, we noted that $\mathsf{Streaming - Filterpd}$ provides speedups of at least $4\times$ over $\mathsf{Filterpd}$ with a smaller memory footprint, in addition to better performance on our metrics due to the streaming approximation.

# 5  EXPERIMENTS

## 5.1  SYNTHETIC EXPERIMENTS - LINEAR REGRESSION

To corroborate our theoretical results, we conduct heavy-tailed linear regression to study the performance of our proposed algorithms.

**Setup**  We generate covariates $x \in \mathbb{R}^p$ from an standard Gaussian distribution. The true regression parameter is set as $\theta^* = [1, 1, \ldots, 1] \in \mathbb{R}^p$. The response $y$ is generated by $y = x^T\theta^* + \sigma w$ where $w$ is drawn from a standardized Pareto distribution with tail-parameter $\beta = 3$. In this setup, we experiment with different $n, p$ and $\delta$. For each setting of $(n, p, \delta)$, and cumulative metrics are reported over 100 trials. We vary $n$ from 100 to 500, $p$ from 20 to 100, $\delta$ from 0.01 to 0.1 and $\sigma$ from 0.01 to 5 on a logarithmic scale.

**Methods**  We compare $\mathsf{RGD - Filterpd}$ with two baselines: Ordinary Least Squares (OLS) and robust gradient descent which uses Algorithm 2 with GMOM as used in Prasad et al. (2020). Note that Prasad et al. (2020) had previously shown that $\mathsf{RGD - GMOM}$ outperformed several other estimators such as Hsu & Sabato (2016) and ridge regression, hence we skip them in our comparison.

**Metric and Hyperparameter Tuning**  For any estimator $\widehat{\theta}_{n,\delta}$, we use $\ell(\widehat{\theta}_{n,\delta}) = \|\widehat{\theta}_{n,\delta} - \theta^*\|_2$ as our primary metric. We also measure the quantile error of the estimator, i.e. $Q_\delta(\widehat{\theta}_{n,\delta}) = \inf\{\alpha : \Pr(\ell(\widehat{\theta}_{n,\delta}) > \alpha) \leq \delta\}$. This can also be thought of as the length of confidence interval for a confidence level of $1 - \delta$. The number of blocks $k$ and the number of iterations $T^*$ is set to $\lceil 3.5 \log(1/\delta) \rceil$ in $\mathsf{RGD - GMOM}$ and $\mathsf{RGD - Filterpd}$ respectively.

**Results**  Figure 3 shows that our $\mathsf{RGD - Filterpd}$ clearly outperforms both baselines. Figures 3a, 3b and 3c are generated with $\sigma = 0.1$. Figure 3a indicates that for any confidence level $1 - \delta$, the length of the oracle confidence interval ($Q_\delta(\widehat{\theta}_{n,\delta})$) for our estimator is better than all baselines. We also see better sample complexity in Figure 3b, and better dimension dependence in Figure 3c. We also observe that when $\sigma$ is increased, which corresponds to a lower signal-to-noise ratio, $\mathsf{RGD - Filterpd}$ performs better in comparison to the other baselines.

We have also conducted synthetic experiments on heavy-tailed mean estimation, and defer the results to the appendix (Section D) due to space constraints. In summary, $\mathsf{Filterpd}$ outperforms both GMOM and the sample mean with respect to the metric considered.

## 5.2  REAL WORLD EXPERIMENTS

With the backing of empirical evidence that the distributions of gradients during training certain generative models are heavy-tailed as shown in Section 3, we seek to apply a heavy-tailed mean estimator over the gradient samples to obtain robust estimates of the mini-batch gradient. Due to the size of the models, it is infeasible to run $\mathsf{Filterpd}$, and hence we use $\mathsf{Streaming - Filterpd}$ instead.

### 5.2.1 GENERATIVE ADVERSARIAL NETWORKS

**Setup** We consider a DCGAN architecture for our experiments, same as the one considered in our investigations presented in Section 3. The model architecture is detailed in the code appendix.

**Methods** We train a DCGAN using Streaming − Filterpd (abbrev. StrFpd) with the optimizer as ADAM for 50000 iterations on the MNIST and CIFAR10 datasets. For comparison, we train other DCGANs with the same initialization, using ADAM; however, the mini-batch gradients for the generator are aggregated via the following methods: sample mean (abbrev. Mean), gradient clipping (abbrev. Clip), removal of gradient samples with largest norm (abbrev. NrmRmv). All other relevant training hyperparameters are detailed in the appendix.

**Metrics** We consider two key metrics: the Inception Score (Salimans et al., 2016) for CIFAR10 and Mode Score (Che et al., 2017) for MNIST, and the Parzen window based log-likelihood estimates as described in Goodfellow et al. (2014).

**Results** We present our results in Table 1. We observe that training with the Streaming − Filterpd provides benefits in terms of high metrics. For CIFAR10, we see a significant improvement in the Inception score. For MNIST, the MODE scores are only slightly higher, however there is a discernible increase in the log-likelihood when trained using Streaming − Filterpd as compared to another baselines.

|  | IS (CIFAR10) | MODE (MNIST) | Parzen LL (MNIST) |
|---|---|---|---|
| StrFpd | **5.20** | **8.94** | **218.95** |
| Mean | 5.07 | 8.91 | 207.61 |
| Clip | 5.09 | 8.92 | 214.29 |
| NrmRmv | 4.94 | 8.87 | 214.25 |
| GMOM | 4.97 | 8.90 | 205.66 |

Table 1: Table of comparison of metrics - inception score (IS), mode score (MODE) and Parzen log-likelihood (Parzen LL)

### 5.2.2 INVERTIBLE FLOW MODELS

**Setup** As done in Section 3, we consider RealNVP – a state-of-the-art model.

**Methods** We train a RealNVP model using Streaming − Filterpd (abbrev. StrFpd) with the optimizer as ADAM for 5000 iterations on the CIFAR10 dataset. For comparison, we train two RealNVP models, both using sample mean for aggregation of gradient samples. However, one of the models does not use *norm clipping* over the estimated gradient, and neither does Streaming − Filterpd. All other relevant training hyperparameters are detailed in the appendix.

**Metrics** We compute the negative log-likelihood on the test set $\text{NLL}_{\text{test}}$ every 250 iterations. We report the test bits-per-dimension (BPD) given by $\text{BPD} = \frac{\text{NLL}_{\text{test}}}{p \cdot \log(2)}$ where $p$ is the dimensionality of the samples. Additionally, to show stability, we report the average test BPD over the last 2000 iterations.

**Results** We present our results in Figure 4 and Table 2. Figure 4 shows the erratic variation in BPD when Streaming − Filterpd is not used and further validates our hypothesis regarding the heavy-tailedness of gradients. Additionally, note that training with Streaming − Filterpd is the most stable among the 3. We also see that Streaming − Filterpd achieves the best average test BPD over the last 2000 iterations.

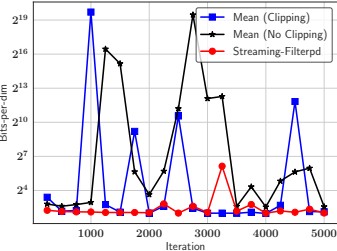

Figure 4: Variation of Test BPD with iterations

| Method | Average Test BPD over last 2000 iterations |
|---|---|
| StrFpd | **12.97** |
| Mean (no-clipping) | 641.66 |
| Mean (with-clipping) | 459.16 |

Table 2: Table of Average Test BPD over the last 2000 iterations

## 6 DISCUSSION

In this work, we study methods for mean estimation, which are applicable for aggregating gradients. This is especially useful when the underlying distribution of these gradients is heavy-tailed, where the sample-mean can be highly sub-optimal. We motivate the need for a robust, heavy-tailed mean estimator by studying distribution of gradients of certain deep generative models. We also develop a heuristic for our mean estimation algorithm that scales to models with millions of parameters. One potential avenue of improvement is to make the heuristic applicable to extremely large models, such as to GANs that generate high quality media or certain language models. Nonetheless, we hope that this work encourages the development of principles approaches to tackling heavy-tailed distributions arising in machine learning.

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
