# OpenReview forum: "Efficient Estimators for Heavy-Tailed Machine Learning"
_ICLR.cc/2021/Conference — Reject_

### Official Review · AnonReviewer3 · 2020-10-25
**This paper seems to make reasonable contribution. I am missing running time in experiments**

**Rating:** 6
**Confidence:** 3

**Review:**

In summary, this paper proposes a novel estimator for the mean estimation of heavy-tailed distribution.

In general, I think this paper makes a good contribution. However, my major concern is that I am missing an important metric for the experiments: the running time. In my view, the authors *must* report the running time of their algorithm and others to make a fair comparison.

In the synthetic experiment, it would be good to include the Streaming - Filterpd algorithm as well so we can compare it with the usual Filterpd, in terms of both accuracy and running time. The author mentioned "at least 4X speed up", but we didn't see the numbers in experiments.

The key part in Algorithm 1 is to discard samples according to their score as in Step 5. It would be good to provide more intuitions for this step. Why the samples with the large score are not good and why discarding them improve the performance? Also, what if we discard a few samples with the largest score instead of randomly select samples? Would that be worse than the current randomized algorithm and why is that?

In theorem 2, what is $d$? I believe it is not the same as the $d$ in Algorithm 3. Is that a similar quantity as $r(\Sigma)$ in equation (3)? Here if $\Sigma = I_p$, then $r(\Sigma) = p$, so the sample complexity is more than p^2 in Theorem 2? It would also be helpful to clearly state that $\theta \in R^p$ at an early stage.

In the line just above Algorithm 1, it is claimed that other methods are impractical because "they have several hyperparameters". This is not convincing. Lasso also has hyperparameters but it does not limit its usage.

In Section 3 when talking about $\alpha$, it might be better to be more quantitative. It is mentioned that the $\alpha$-index is close to 2 when samples are close to Gaussian. How close? I think 1.85 as in Figure 2(a) and 2(b) is also close to 2.

Step 2 in Algorithm 2: the "$log(\delta/T)$" should be a typo, is that "$log(T/\delta)$" instead?

---

> ### Author Response · Authors · 2020-11-22
> **Author Response**
>
> Thank you for your review. We address your queries below.
>
> Q: **The key part in Algorithm 1 is to discard samples according to their score as in Step 5. It would be good to provide more intuitions for this step.**
>
> A: The algorithm works because of a key insight relating the scores computed. If there are samples that are far away from the true mean, then the mass (i.e., the score) placed on these samples is higher than the mass placed on the samples closer to the mean. This means that the probability of discarding a point that is a far away from the mean is higher. In the proof of Lemma 3 and 4, we precisely work with this intuition.
>
> Q: **Also, what if we discard a few samples with the largest score instead of randomly select samples?...**
>
> A: That is an astute point. The focus on the randomized version was due to the nuanced technical arguments needed for a high-probability bound analysis. In the appendix where we give the results for mean estimation, we also compare the randomized version and the $\mathrm{argmax}$ version, and find that there is not much variation in the results. Accordingly, in our default implementation of this algorithm, we do use the $\mathrm{argmax}$ variant as suggested by the reviewer.
>
> Q: **...it is claimed that other methods are impractical because "they have several hyperparameters". This is not convincing.**
>
> A: We would like to state that the method referred to in that statement was specifically the algorithm due to Lei et al (2019), the authors of this paper haven’t conducted experiments in their work, and we had to implement their algorithm from scratch, and present preliminary results for their estimator (Table 3, Appendix D), and note that this doesn’t perform as well as our proposed method. Their algorithm is based on deriving an approximation of the SDP approach in Cherapanamjeri et al (2019). However, while theoretically optimal, their algorithm relied on various constants that were loosely bounded, and which thus required tuning for a task as simple as mean estimation. Some instances are: (a) their theoretical results (Theorem 1.1) requires the number of groups to be $C \log(1 / \delta)$, where $C$ is a large constant; (b) the number of steps to run the distance and gradient estimation functions (Algorithm A.1 and A.2) are effectively unknown due to constants being hidden in the $O(.)$ notation. On the other hand, our algorithm Filterpd has one tuning parameter which is $T^{*}$.
>
> Q: **...when talking about $\alpha$, it might be better to be more quantitative…**
> A: Theoretically, the heavy-tailedness of $\alpha$-stable distribution is qualitatively unsmooth. By this, we mean that even for $\alpha = 2 - \epsilon, \epsilon > 0$, the distribution can become extremely heavy-tailed (see figure 1.d for example). Furthermore, we repeatedly use the estimator to compute the $\alpha$-index for samples from a Gaussian distribution, and note that for the same sample size as those in Figure 2 $(n=10000)$, we don’t see significant deviations from the theoretical value of 2. To be precise, we obtain a 99% confidence interval of $(1.99,2.03)$ for Gaussian data. Therefore, 1.85 still indicates that the distributions are heavy-tailed.
>
> Furthermore, we would like to state that the alpha-index estimator is not a precise way to measure heavy-tailedness due to the assumptions required by the estimator such as symmetricity, and is more indicative in nature.

---

### Official Review · AnonReviewer4 · 2020-10-27
**Interesting experiments but sub-optimal theoretical results**

**Rating:** 5
**Confidence:** 5

**Review:**



## Summary

The paper studies the problem of high-probability mean estimation for heavy-tailed distributions, i.e., constructing a high-probability confidence intervals for the mean, when the underlying distribution has only finite low-degree moments.  The paper motivates this problem from the view-point of machine learning algorithms, where the gradients are heavy-tailed. 	Especially in deep generative networks, the paper  highlights the heavy-tailed nature of gradients via experiments. On a theoretical side, the paper derives bounds for mean estimation when the distribution has bounded fourth-moments.

---
## Pros

+ The paper highlights the heavy-tailed nature of gradients in the intermediate stage of training deep generative models via experiments.
+ The paper derives theoretical bounds for their mean estimation algorithm, and then applies it to study generalized models via gradient aggregation.
+ A streaming heuristic of Algorithm 1 is proposed that has promising results in experiments.

---
## Cons

The key concern is that despite several claims of "near-optimality" in the paper, the theoretical results in the paper are significantly lacking in the sample complexity (I understand the rates are subgaussian after this stringent sample complexity is satisfied). Please see below for detailed comments on mean estimation and linear regression.

**Mean estimation**
+ Theorem 1 requires the sample complexity of $n \geq r^2(\Sigma) \log(1/\delta)$ (and a bounded 4-th moment condition), whereas the "true subgaussian bound" has a sample complexity of only $n \geq \log(1/\delta)$ without any higher moments.
+ This difference is considerable, especially, in high-dimensions, which is the focus of this work. For example, a typical example is when $r(\Sigma) = p$ and $\delta = 2^{-p}$ (high-probability confidence typically required for union bound over covers), Theorem 1 requires $n \geq p^3$, whereas the information-theoretically optimal is only $n \geq p$.
- As a result, the theoretical results in Theorem 1 are not even applicable in several interesting modern regimes, including the experiments in this paper. I am thus skeptical of the recurring claim of near-optimality of Theorem 1 in the paper (abstract, Section 2 heading, etc. ).
- As mentioned in Appendix, without any higher-moments assumption, the rates in Theorem 1 are worse than GMOM. A more thorough comparison of Theorem 1 with GMOM should be added (preferably in the main text). For example, even though the error in GMOM is large, the sample complexity is dimension-independent.
- The results should also be compared with the paper Diakonikolas, Kane, Pensia (see below), where the algorithm doesn't seem to require more hyperparameters than Algorithm 1 (which is claimed to be as the primary shortcoming of prior works and motivation for Theorem 1).

**Linear Regression**:
- The aforementioned drawbacks in Theorem 1 are also reflected in Theorem 2, when the mean-estimation sub-routine is applied to linear regression.Again, I believe that the claim of "near-optimality" is somewhat misleading. In the same parameter regime of \delta = 2^{-p}, the sample complexity in Theorem 2 is at least $n \geq p^3$, compared to the optimal of $n \geq p$ given in Lugosi and Mendelson, 2016 (see below for reference).
- Cherapanamjeri et al. (2019b) (CHKRT) is discussed in Appendix B.2, but I believe their results have been misrepresented. Again considering the parameter regime of $\delta = 2^{-p}$, the sample complexity of Theorem 2 is at least $n\geq p^3$, which is incorrectly stated in Appendix as $p^2$ (Please let me know if I miscalculated the sample complexity of Theorem 2).
- In general, the sample complexity in [CHKRT, Theorem 5.1] is $p\sqrt{\log(1/ \delta)}$ as compared to $p^2 \log(1/\delta)$ in Theorem 2. Thus I believe the claim of "best known result" in the remark after Theorem 2 is incorrect (see also Depersion, 2020).
- Moreover, the sample complexity in Thm. 2 increases with $T$, and $T$ depends on $|| \theta^* ||$ (which has been hidden in $O(.)$ notation in Appendix B.2).  This is in stark contrast with results in CHKRT, where only the time complexity (and not sample complexity) depends on $||\theta^*||$. Actually, they use OLS to warm-start their algorithm, reducing their running time to linear independent of $||\theta^*||$. But doing the same here, would further increase the sample-complexity in Theorem 2.
- Theorem 2 should also be compared with the paper Depersin (2020) (see below), preferably, in the main paper.

---
## Score

Overall, I vote for rejection. The current theoretical results (Theorem 1 and Theorem 2) have significantly large sample complexity, and do not explain the experimental results of the paper ($n = O(p)$ and $ r(\Sigma) = p$ in experiments). The significance of these shortcomings are also not discussed in the main text, where the results are repeatedly claimed as near-optimal. I liked the experiments in the paper but they are currently not extensive for an acceptance on its own. Perhaps authors can focus more on the experiments in a resubmission.


---
### Other major comments

**Streaming algorithm, Section 4**

+ *Moreover, it is not viable to run* $T^*$ *leading eigenvector computations for models with millions of parameters, since the time complexity of this operation is proportional to* $p$.
I am confused by this statement: even the simple baseline of sample mean would have time complexity of np. This line should be clarified.
+ *Recall that in Filterpd, we require to compute 1)* $ T^*$ *covariances and 2)* $T^*$ *leading eigenpairs*
This line implies that the filter algorithm needs to calculate these covariance matrices explicitly. As the prior work as shown (see, for e.g., Lei et al. (2019) and Depersin and Lecue (2019)), a simple power iteration can be used which only requires dot products (achievable $O(np)$ time). This is a standard procedure in this literature.
+ In experiments with Streaming-Filterpd, the runtime (wall-clock time) should also be reported with respect to baselines (Mean, Clip, NrmRmv).
+ Seeing that the performance of GMOM closely matches the performance of Filterpd in Figure 3 and simplicity of computing GMOM, I would like to see the comparison of GMOM with Streaming-Filterpd in Table 1 and Figure 4.


---
### Other minor comments

+ The claims of near-optimality should be replaced with something objective.
+ For an $\alpha^*$-heavy-tailed distribution, the quantity $E X^{\alpha}$ only exists if $\alpha$ is "strictly" less than $\alpha^*$. The equality should be removed from the paragraph below Definition 1.
+ The citation to Minsker (2015) before Eq. (2) should be modified: in the current form, the text suggests that the Minsker (2015) showed a lower bound on the sub-optimality of GMOM.
+ I am not sure if citation to Catoni and Giulini (2007) is correct. The bounds in that paper were not *truly subgaussian* as the results depended on the raw moments of the distribution.
+ The paper uses $d$ and $p$ interchangeably. For example, Eq. (5) has both, but only $p$ is defined.
+ It is a bit confusing to use T in Algorithm 3, where T is the iteration number of optimization algorithm vs T^* in Algorithm 1 which is the parameter of sub-routine.
+ Figure 3 (c): It seems that the dependence on $p$ is linear as opposed to $\sqrt{p}$. Is there a reason for this?
+ Section 4: "This eigenpair is not required to be recomputed the current iteration"
+ Section 5: "For comparison, we other DCGANs with the same initialization"
+ as -> are on Page 2: "This is because sample mean estimates as ..."

---
### Other References:

Ilias Diakonikolas, Daniel Kane, Ankit Pensia. Outlier Robust Mean Estimation with Subgaussian Rates via Stability. 2020.
Jules Depersin. A spectral algorithm for robust regression with subgaussian rates. 2020.
Gabor Lugosi and Shahar Mendelson. Risk Minimization by Median-of-Means Tournaments. 2016.
There are other recent works for both mean estimation and linear regression, but they were made public after the ICLR guideline of Aug 2, 2020.

---

> ### Author Response · Authors · 2020-11-22
> **Author Response**
>
> We thank the reviewer for their extremely detailed review and for noting that the experiments are interesting.
>
> Q: **The theoretical results in the paper are significantly lacking in the sample complexity (I understand the rates are subgaussian after this stringent sample complexity is satisfied). Theorem 1 requires $n \geq p^{3}$ whereas the information-theoretically optimal is only $n\geq p$.. , the sample complexity of Theorem 2 is at least $n \geq p^{3}$..**
>
> A: We note that your calculations use a worst case bound on $r(\Sigma)$. We note that for practical problems, $r(\Sigma)$ scales closer to a constant, in which case our sample complexity is $O(p)$.
>
> Moreoever, we would like to note we derive high-probability vs in expectation bounds as in much of recent work. For the former, we used state-of-the-art martingale analysis tools available at the time of our original preprint.
>
> Lastly, we note that the $n \geq C r^2(\Sigma) \frac{\log^2(\frac{p}{\delta})}{\log(\frac{1}{\delta})}$ is not required.
> The actual rate of the estimator, as can be seen from the proofs, is:
>
> $$\begin{eqnarray}
> \mathrm{rate} &=&OPT_{n, \Sigma, \delta} \\\\
> &+& C_{1}r(\Sigma)^{\frac{1}{4}}\left(\frac{1}{n}\right)^{\frac{1}{8}} \frac{\log^{\frac{1}{4}}\left(\frac{p}{\delta}\right)}{\log^{\frac{1}{8}}\left(\frac{1}{\delta}\right)} \sqrt{\frac{1}{n}\log\left(\frac{1}{\delta}\right)} \\\\
> &+& C_{2}(r(\Sigma)||\Sigma||_{2})^{\frac{1}{2}}\left(\frac{1}{n}\right)^{\frac{1}{4}} \frac{\log^{\frac{1}{2}}\left(\frac{p}{\delta}\right)}{\log^{\frac{1}{4}}\left(\frac{1}{\delta}\right)} \sqrt{\frac{1}{n}\log\left(\frac{1}{\delta}\right)} \\
> \end{eqnarray}$$
>
> $$\begin{eqnarray}
> \\phantom{\\mathrm{rate}}&+& C_{3}(r(\\Sigma)||\\Sigma||_{2})^{\\frac{1}{2}}\left(\\frac{1}{n}\\log\\left(\\frac{1}{\\delta}\\right)\\right)^{\\frac{3}{4}}
> \end{eqnarray}
> $$
>
> Observe that the second term decays slower than the information theoretic lower bound rate (OPT), however, when,  $n \geq C r^2(\Sigma) \frac{\log^2(\frac{p}{\delta})}{\log(\frac{1}{\delta})}$, we obtain the optimal sub-Gaussian rate.
>
> With regard to linear regression, as stated in the general comment, several recent works make the assumption that the covariance is identity or is known, which is not practically feasible either. We don’t rely on such assumptions, and give results for general unknown bounded covariances, and stress that ***these are the only such results for the unknown covariance case***.
>
> Q: **A more thorough comparison of Theorem 1 with GMOM should be added (preferably in the main text).**
>
> A: We added the comparison to GMOM in the appendix due to space constraints.
>
> Q: **The results should also be compared with the paper Diakonikolas, Kane, Pensia (DKP) (see below)**
>
> A: We thank the reviewer for pointing out this recent work. Note that the DKP only studies the problem of mean estimation and shows that for the algorithm proposed in our paper, one can obtain better bounds than the one that we give. However, we were the first to note the relevance of a SVD based method for mean estimation. As a side note, we would like to highlight that DKP cites the arXiv version of this work.
>
> Q: **In general, the sample complexity in [CHKRT, Theorem 5.1] is $p\sqrt{\log⁡(1/\delta)}$ as compared to $p^{2}\log⁡(1/\delta)$ in Theorem 2...**
>
> A: The guarantees in CHKRT for linear regression are only given in the case where the covariance is identity, whereas our analysis holds for general unknown covariances. The gap between known and unknown covariance is actually a deeper technical issue, as is pointed out by CHKRT as a limitation of their work (Footnote 2 ,Page 5, arXiv version). Moreover, we would like to note that we were able to attain these results for linear regression and GLMs by merely changing the underlying mean estimator in the framework of Prasad et al (2020).
>
> Q: **Comparison to Depersin (2020)**
>
> A: The primary limitations of Depersin (2020) is that they assume that the covariance is known and make note of this in the conclusion of the paper. We would like to reiterate that our theoretical results hold for general bounded unknown covariances. Hsu and Sabato (2016) is the most recent work that doesn’t require the assumption of known covariance, and we achieve better rates than those presented in their paper.
>
> Q: **I am confused by this statement: even the simple baseline of sample mean would have time complexity of $np$...**
>
> A: We apologize for this error, the correct statement should have been "...since the additional time complexity of this operation is proportional to p and is not as efficient as computing the sample mean." and this has been changed in the current version of the draft.

---

> > ### Author Response · Authors · 2020-11-22
> > **Author response continued**
> >
> > Q: **...a simple power iteration can be used which only requires dot products (achievable $O(np)$ time)...**
> >
> > A: Certainly, we are aware of this. In fact, our implementation of Algorithm 1 for the synthetic results used power-iteration based methods to compute the leading eigenvector. What we intended to convey was that running $T^{*}$ independent power-iterations is not feasible to estimate the mean of an extremely high-dimensional vector such as gradients of loss functions with respect to the parameters of a large neural network. We have made this clear in the current version of the draft. However, the constant in the $O(.)$ is typically large for power iteration, and therefore is still expensive.
> >
> > Q: **I would like to see the comparison of GMOM with $\mathsf{Streaming-Filterpd}$**
> >
> > A: We have added the comparison to GMOM in the main text - Table 1. We observe that GMOM does not perform as well as $\mathsf{Streaming - Filterpd}$.
> >
> > Q: **Regarding minor comments**
> >
> > A: Thank you for your attention to detail. We have changed the existence of moments definition which was a  typographical error, the statement above Equation 2. Regarding the variation with p in Figure 3c, please note that the y-axis is the length of the confidence interval instead of the actual error.

---

### Official Review · AnonReviewer1 · 2020-10-27

**Rating:** 6
**Confidence:** 4

**Review:**

I thank the authors for their detailed reviews. I have updated my score

---

The submission presents a robust estimator for the mean of heavy-tailed distributions for application to neural network training. Understanding and dealing with the distribution of the noise in stochastic optimization for machine learning is relevant to both practical and theoretical aspects of machine learning and relevant to the ICLR community.

The main weakness of the submission is a lack of clarity in the contributions and presentation. Key statements are vague, basic notation is not defined and the writing and supporting figures would benefit from an additional pass. More details below.

My initial recommending is towards a rejection. I think the paper needs major revisions to improve the presentation rather than a set of minor improvements that are easily fixed. But I am open to increase my score depending on the results of the discussion period if the issues below are addressed.

**Major issues**

* Unclear theoretical contributions.

  It is unclear from Section 2 whether the proposed algorithm and Theorem 1 are novel theoretical contributions or $\epsilon$-modification to existing work, such as the works of Diakonikolas et al. and Lugosi et al., that is more amenable to. Either are valuable contributions, but this needs to be made clear. If Thm. 1 is a significant theoretical contribution, the theoretical novely needs to be expanded upon and the differences with existing work made explicit. If the results are straightforward from existing theory but Alg. 1 more convenient for applications, the issues with existing estimators need to be clearly highlighted

* Basic notation is not defined. I could only infer the following by reading referenced material

  * the dimensionality $p$ (p.2 or Eq. 3)
  * the unit ball $\mathcal{S}^p$ (p. 2)
  * an outer product $(v)^{\otimes 2}$ (Alg. 1)
  * a constraint set $\Theta$ (Alg. 2)
  * the constant $d^2$ (Thm. 2, Eq. 5) is still mysterious

* Clarity about the heuristic nature of Alg. 3.

  There is nothing wrong with simplifications to allow the method to scale to large datasets, even if at the cost of some rigor. But the section needs to make clear that the simplifications are heuristic in nature, and not attempt to cover it with technical but wrong language. Eg: "reusing previous gradient samples to improve concentration" (p. 6) is inaccurate as the samples are not independent.
"it is unreasonable to expect that [the distribution of gradients] are vastly different in most cases, due to the smoothness of the objective" (p. 6) Reasonability is subjective, and the presented argument is wrong. Neural network objectives are most often non-smooth. This holds whether smoothness refers to differentiability, due to ReLU activations, or the Lipschitzness of the function or the gradient due to multiple layers.

**Minor comments:**

* The figures need additional work as they are currently unreadable when printed.
* The writing needs improvement, as some sentences are incomplete or contain duplicated words. Eg "This eigenpair is not required to be recomputed the current iteration" (p. 6), "sufficiently large enough" (remarks), "achieves the the optimal sub-Gaussian" (remarks)
* Some of the cited preprints have been published (eg Che et al., Cherapanamjeri et al.). Please make sure your references are accurate and up-to-date.

---

> ### Author Response · Authors · 2020-11-22
> **Author Response**
>
> Thank you for your detailed response, and we apologise for the missing definitions of terms and fonts that have hindered a smooth reading.
> We would address the issues raised in order.
>
> Q. **It is unclear from Section 2 whether the proposed algorithm and Theorem 1 are novel theoretical contributions or $\epsilon$ -modification to existing work, such as the works of Diakonikolas et al (2017)….**
>
> A. As noted in the statement above, we would first like to note that our arXiv manuscript is concurrent to or predates several ongoing work in this area.
>
> We note three key distinctions of our results in comparison to Diakonikolas et al (2017):
> We obtain a high-probability bound rather than a bound that holds in expectation, which requires a more nuanced analysis of the probability process (via sub-martingales). In particular, due to the key distinction, one *cannot* read our bounds off from their analysis: it requires further nuanced technical arguments.
> The analysis of a subset of mentioned recent work is targeted to the Huber contamination setting (where a portion of the data is arbitrarily corrupted), whereas we consider the heavy tailed setting. On a related note, their results are mainly focused on the isotropic setting ($\Sigma \preceq \sigma I_p$), while our lemma carefully tracks the dependence on the covariance and gets tight dependence on problem parameters such as $\mathrm{trace}(\Sigma)$, which is a key facet in heavy-tailed analysis.
> Finally, they don't observe the better rates for 4th moment bounded distributions beyond what one can get if they relax bounded 2nd to 4th moment.
>
> In gist, we extend their analysis in natural if non-trivial directions which are critical to obtain high-probability bounds for heavy tailed estimation: which is the core goal of the paper.
> We will be sure to add a more nuanced discussion of these technical arguments, as suggested by the reviewer.
>
> Q. **Clarity about the heuristic nature of Alg. 3.**
>
> A. We would first like to note that we have emphasized from the outset that the algorithm presented is a heuristic, which is evidenced in the title of the section itself. We would like to clarify that the statements presented in the paragraph referenced are meant to provide intuition, and are not exactly precise as pointed out. To that effect, we have altered or removed incorrect and misleading wording in the current version of the draft.
>
> Q. **Some of the cited preprints have been published...**
>
> A. Thank you for pointing these out again, we have made the changes in the current version of the draft, by including latest versions of as many references as possible.

---

### Official Review · AnonReviewer2 · 2020-10-28
**A good paper and marginally above acceptance threshold**

**Rating:** 6
**Confidence:** 4

**Review:**

The author(s) propose a computationally efficient mean estimator for generative distribution that are "heavy-tailed" in nature. The phenomenon of heavy tailed distributions for gradients in the training stage of generative models are common in nature and the proposed method aims to alleviate this problem by constructing a robust gradient estimator in such situation. The proposed methodology is well backed up by synthetic and real data examples. The topic is interesting and the proposed methodology is novel.

I have the following queries for the author(s)

-In the definition of bounded 2K-moments why the vector v needs to be in a (p-1) dimensional sphere? Is it necessary for the vector to be rotationally invariant?

-Along the lines of the first query, does the definition of bounded 2K-moment has any relation with sub-gaussian moment condition?

-In the streaming filterpd algorithm (Algorithm 3) how robust is the rank-1 approximation of C_{t-1} by the leading eigen-pair?

-I am a bit skeptical with the bounded 8th moment condition in the linear model example?

-In theorem 2 do you allow dimension p to grow or is the result not tenable for the high dimensional case?

- It would be interesting to see an experiment between RGD-GMOM and RGD-Filterpd under different signal-to-noise ratio for heavy tailed mean estimation in the linear regression setting.

---

> ### Author Response · Authors · 2020-11-22
> **Author Response**
>
> Thank you for your review. We give answers for your queries below.
>
> Q. **It would be interesting to see an experiment between $\mathsf{RGD-GMOM}$ and $\mathsf{RGD-Filterpd}$ under different signal-to-noise ratio for heavy tailed mean estimation in the linear regression setting.**
>
> A. In the revised draft, we show our new results in Figure 3d. In short, we see that for increasing noise, the performances of the estimators worsen, yet $\mathsf{RGD-Filterpd}$ performs better than the baselines, which is desirable.
>
> Q. **In the definition of bounded 2K-moments why the vector $v$ needs to be in a $(p-1)$ dimensional sphere? Is it necessary for the vector to be rotationally invariant?**
>
> A. This is a commonly used definition. Intuitively, this states that the projection of the centred random variable along any direction is well behaved. Consider a random variable $X = (X_1, X_2)$, where $X_1$ is an exponential random variable and $X_2$ is a Pareto random variable with parameter 2 that are independent. Now, for $v = (0, 1), \mathbb{E}[(v^{T}(X))^{4}] = \infty$, and therefore, doesn’t satisfy the condition of bounded 2K-moments for k = 2.
>
> Q. **does the definition of bounded 2K-moment has any relation with sub-gaussian moment condition?**
>
> A. Using the well-known fact about the moments of a sub-Gaussian random variables, which is that $\mathbb{E}[X^{2k}] = O((2k)^{2k})$ - we can see that for sufficient large constants $C_{2k}$ that sub-Gaussian random variables satisfy the bounded 2K-moments definition for all $k \geq 1$. Therefore, the sub-Gaussian moment condition is a stronger condition than the bounded 2k-moments condition.
>
> Q. **In the streaming filterpd algorithm (Algorithm 3) how robust is the rank-1 approximation of $C_{t-1}$ by the leading eigen-pair?**
>
> A. We are not exactly sure about what the reviewer had in mind w.r.t. robustness, but we note that this is a standard approximation for computational considerations, even if we do not explicitly have a worst case bound on the approximation gap i.e., $||A - A^{(1)}||_{F}$.
>
> Q. **I am a bit skeptical with the bounded 8th moment condition in the linear model example?**
>
> A. This is an assumption previously made in Prasad et al (2020), whose framework we borrow for presenting guarantees on our estimator applied to linear regression. We present a comparison to other existing rates in the literature, and remark that we improve on the previously known best rates for heavy-tailed linear regression (Hsu and Sabato (2016) and Prasad et al (2020)) by merely changing the heavy-tailed mean estimator.
>
> Q. **...do you allow dimension p to grow or is the result not tenable for the high dimensional case?**
>
> A. There are two facets to this question. First, our guarantees allow $p$ to grow with $n$ as long as $\frac{p}{n} \to 0$. However, in the case when $\frac{p}{n} \to \infty$$ (i.e. $$n \ll p$) - which is popularly known as the high dimensional regime, one would need an additional sparsity constraint.

---

> > ### Comment · AnonReviewer2 · 2020-11-24
> > **Reasonable response but some doubts still lingering**
> >
> > Overall the author(s) response are reasonable but few questions still remain
> >
> > - I understand the rank-1 approximations are computationally standard however one should have some remark about how good the approximation is even if one does not have the worst case bound
> >
> > - some heuristic about high dimensional regimes would be good
> >
> > -Happy with the simulation results on RGD-GMOM and RGD-Filterpd
> >
> > - I understand the bounded 8th moment condition occurrence in the literature earlier but how much extra technical contribution the author(s) provide in terms of novelty in theoretical results with that assumption is not quite clear. few other reviewers had the same query and going through the author(s) response I am not entirely clear about the explanations.
> >
> > I still keep my previous rating of "marginally above acceptance threshold"

---

> > > ### Author Response · Authors · 2020-11-24
> > > **Thanks for going through the response.**
> > >
> > > We address your questions below:
> > >
> > > Q: **I understand the rank-1 approximations are computationally standard however one should have some remark about how good the approximation is even if one does not have the worst case bound**
> > >
> > > A: In general, rank-1 approximations are good when you have a exponential tail for the eigenvalues of the matrix. In our case, we observe that as iterations proceed, the gap between the top eigenvalue and the second highest eigenvalue increases, but not drastically. This indicates that we could perform better by considering a higher rank approximation, but will be more expensive. Despite this, our experimental results have shown that a rank-1 approximation works well.
> > >
> > > Q: **some heuristic about high dimensional regimes would be good**
> > >
> > > A: Perhaps in the high dimensional case, one could consider the proximal mapping for the update on line 6 of Algorithm 2 by adding a $\ell_{1}$-norm penalty. We have not considered this in our work.
> > >
> > > Q: **I understand the bounded 8th moment condition occurrence in the literature earlier but how much extra technical contribution the author(s) provide in terms of novelty in theoretical results with that assumption is not quite clear. few other reviewers had the same query and going through the author(s) response I am not entirely clear about the explanations.**
> > >
> > > A: We would like to direct your attention to the general response above (https://openreview.net/forum?id=5K8ZG9twKY&noteId=A6dVXQg7qUR), where we have clarified the core theoretical contributions of the work. In short, with regard to linear regression (and GLMs), we have also provided theoretical results and the only known algorithms for linear regression and GLMs for the case where the distribution of covariates has an unknown covariance at the cost of some minor assumptions. For the unknown covariance case, the work by Hsu and Sabato (2016) -- who worked with similar assumptions -- had the previous best results for linear regression in the unknown covariance case, which we improve upon.

---

### Author Response · Authors · 2020-11-22
**Uploaded a revised draft with clarifications and some new experiments**

We thank the reviewers for their interest in this work and for providing valuable feedback towards improving this work.

Most of the reviewers have found the empirical findings and the heuristic scalable approximation to our main algorithm interesting; these are indeed the core contribution of the work, and we thank them for their assessment.

Regarding our theoretical contributions relative to some recent work: we would like to note that the arXiv version of this work (posted 1 July 2019) has predated or has been concurrent to these other papers (this slew of ongoing work indicates the importance of the problem). At the time our preprint was completed and uploaded to arXiv, our results were the best-known for computationally-efficient and practically implementable procedures, improving on the geometric median with respect to rates, and in particular, was the ***first*** work to connect the two seemingly disparate notions of robustness: heavy-tailedness and adversarial contamination.
However, these theoretical results are the focus of a subsection (Section 2 (mean estimation)) of the current manuscript only.

We have also provided theoretical results and the ***only*** known algorithms for linear regression and generalized linear models for the case where the distribution of covariates has an unknown covariance. In particular, Cherapanamjeri et al (2020) and Depersin (2020) get better theoretical rates for linear regression under the restrictive and impractical assumption of known and/or identity covariance. Moreover, there are reasons to believe that SDP based algorithms such as Cherapanamjeri et al (2020) don’t work for the general unknown covariance case.

Nonetheless, we now acknowledge these concurrent or later results, and remove claims of near-optimality in the draft where appropriate. While these are outdated, yet novel theoretical contributions, we would like to point out that this is only a section of the paper. A core contribution is the identification of heavy-tailed gradients arising during the training of certain popular deep generative models, and application of a heuristic variant of the proposed heavy-tailed estimators to the task of training these models, and no previous work on heavy-tailed estimation has made this connection in the recent past.

We now address some common queries raised by the reviewers.

Regarding the runtime for our algorithms: we present this in Section E.1 of the Appendix (Table 4). We note that the first estimator presented in the paper is time-consuming; but it is nonetheless a more practical and easier-to-implement estimator than the other well-known, polynomial time estimators in the area of heavy-tailed estimation, and where the latter are far from being used at the scale of our experiments. We note moreover that we propose a heuristic approach that works well.

We also thank the reviewers for identifying unintended notational inconsistencies and typographical errors, and we have fixed them in the current version of the draft and apologize for the inconvenience. Reviewer1 also brought up the issue of smaller image sizes, which we have now scaled and have additionally added descriptions of the legend / axes in the caption of the figures.

We answer reviewer specific questions via replies below.

---

### Decision · Program_Chairs · 2021-01-07
**Final Decision**

**Decision:**

Reject

**Comment:**

This paper studies the problem of multivariate mean estimation with a focus on the heavy-tailed setting.
The authors give an algorithm for this estimation task and then use it (in essentially a black-box manner) to obtain heavy-tailed
estimators for various supervised learning tasks. As pointed out by one of the reviewers and my own reading, the theoretical
contributions of the paper are weak and are subsumed by related work (some of which is not cited in the submission).
More generally, the extensive recent literature on the topic is not accurately represented in both the submission itself and the response to the reviewers comments. On the other hand, the experimental results of the paper hold some promise. However, at this stage, these experimental contributions by themselves are in my opinion insufficient to merit acceptance.